# The Use of Heterologous Antigens for Biopanning Enables the Selection of Broadly Neutralizing Nanobodies Against SARS-CoV-2

**DOI:** 10.3390/antib14010023

**Published:** 2025-03-07

**Authors:** Vazirbek S. Aripov, Anna V. Zaykovskaya, Ludmila V. Mechetina, Alexander M. Najakshin, Alexander A. Bondar, Sergey G. Arkhipov, Egor A. Mustaev, Margarita G. Ilyina, Sophia S. Borisevich, Alexander A. Ilyichev, Valentina S. Nesmeyanova, Anastasia A. Isaeva, Ekaterina A. Volosnikova, Dmitry N. Shcherbakov, Natalia V. Volkova

**Affiliations:** 1State Research Center of Virology and Biotechnology VECTOR, Rospotrebnadzor, Koltsovo 630559, Russiailyichev@vector.nsc.ru (A.A.I.); dnshcherbakov@gmail.com (D.N.S.); volkova_nv@vector.nsc.ru (N.V.V.); 2Institute of Molecular and Cellular Biology, Siberian Branch of Russian Academy of Sciences, Novosibirsk 630090, Russia; 3Genomics Core Facility, Institute of Chemical Biology and Fundamental Medicine Siberian Branch of Russian Academy of Sciences, Novosibirsk 630090, Russia; 4Synchrotron Radiation Facility—Siberian Circular Photon Source “SKlF” Boreskov Institute of Catalysis of Siberian Branch of the Russian Academy of Sciences, Koltsovo 630559, Russia; arksergey@gmail.com (S.G.A.);; 5Department of Natural Sciences, Novosibirsk State University, Novosibirsk 630090, Russia

**Keywords:** nanobodies, single-domain antibodies, VHH, recombinant antibodies, SARS-CoV-2, phage library, affinity selection, biopanning, AlphaFold, protein–protein docking, molecular dynamics

## Abstract

**Background:** Since the emergence of SARS-CoV-2 in the human population, the virus genome has undergone numerous mutations, enabling it to enhance transmissibility and evade acquired immunity. As a result of these mutations, most monoclonal neutralizing antibodies have lost their efficacy, as they are unable to neutralize new variants. Antibodies that neutralize a broad range of SARS-CoV-2 variants are of significant value in combating both current and potential future variants, making the identification and development of such antibodies an ongoing critical goal. This study discusses the strategy of using heterologous antigens in biopanning rounds. **Methods:** After four rounds of biopanning, nanobody variants were selected from a phage display library. Immunochemical methods were used to evaluate their specificity to the S protein of various SARS-CoV-2 variants, as well as to determine their competitive ability against ACE2. Viral neutralization activity was analyzed. A three-dimensional model of nanobody interaction with RBD was constructed. **Results:** Four nanobodies were obtained that specifically bind to the receptor-binding domain (RBD) of the SARS-CoV-2 spike glycoprotein and exhibit neutralizing activity against various SARS-CoV-2 strains. **Conclusions:** The study demonstrates that performing several rounds of biopanning with heterologous antigens allows the selection of nanobodies with a broad reactivity spectrum. However, the fourth round of biopanning does not lead to the identification of nanobodies with improved characteristics.

## 1. Introduction

Nanobodies, also known as single-domain antibodies or VHH, are variable domains of special non-canonical camelid antibodies (HCAb, heavy-chain antibodies), consisting solely of truncated heavy chains and completely lacking light chains. Nanobodies possess several unique properties, such as high solubility, resistance to low pH, and conformational stability across a wide temperature range [1]. These molecules are relatively small (~15 kDa) and are capable of effectively binding to “hidden” conformational epitopes of antigens that are inaccessible to full-length antibodies. To date, numerous nanobody variants targeting viral antigens have been generated from a variety of libraries, including naïve, immune (using immunized animals such as alpacas, llamas, and camels), and synthetic libraries [2,3,4,5,6,7,8,9,10,11]. Bacteriophages or yeast can be used for nanobody repertoire display, while large quantities can be produced using suitable expression systems, including *Escherichia coli*, yeast, plant cells, and mammalian cells [12,13,14].

The main target of protective antibodies is the surface spike (S) protein, which mediates viral infection and pathogenesis. It is responsible for binding to receptors, subsequent membrane fusion, and viral entry into the host cell [15,16,17]. The high degree of glycosylation makes this protein a challenging target for neutralization by classical antibodies [18]. However, recognition of antigenic determinants on this protein is possible with nanobodies. Since the onset of the COVID-19 pandemic, a series of studies have been published on the generation of nanobodies against various strains of SARS-CoV-2 [14,19,20,21]. Due to the high variability in the amino acid composition of the S protein, previously obtained antibodies, including nanobodies, are unable to neutralize new strains and lose their relevance [22,23]. A potential solution to this problem may be nanobodies targeting conserved regions, capable of neutralizing a broad spectrum of SARS-CoV-2 strains [24,25]. In the case of this virus, the pace of obtaining and studying antibody variants using the classical phage display approach, which involves creating an immune library and performing biopanning rounds, lags behind the evolution of the virus. However, changing the biopanning strategy, including the number of rounds and the choice of target antigens, may ensure the identification of necessary variants even from a narrow diversity of naive or immune libraries obtained by immunizing animals with a single antigen.

Given the rapid evolution of SARS-CoV-2, researchers have developed various immunization and biopanning strategies aimed at selecting broadly neutralizing nanobodies. These approaches differ in the immunogens, antigens used, and the number of selection rounds, which influence the diversity and specificity of the nanobodies obtained. In the study [26], a phage display library was used to select nanobodies, generated after three immunizations of an alpaca with recombinant SARS-CoV-2 RBD. Two rounds of biopanning with RBD led to the enrichment of the library with RBD-specific nanobodies, and the titre of the eluate obtained was 2 × 10^5^ CFU/mL (colony-forming units per Milliliter). ELISA analysis of 62 phage clones selected after two rounds of biopanning (31 after each round) showed that 19 phage clones (61%) after the first round and 30 (97%) after the second round specifically interact with RBD. Among the three nanobodies (aRBD-2, aRBD-5, and aRBD-7) that demonstrated neutralizing activity against SARS-CoV-2, aRBD-2 was the most effective. In a later article by the same author [27], the crystal structures of these nanobodies with RBD were discussed, showing that aRBD-2 binds to highly conserved regions and maintains binding with RBD variants Alpha, Beta, Gamma, Delta, Delta Plus, Kappa, and Lambda, and Omicron BA.1 and BA.2 subvariants. In contrast, aRBD-5 and aRBD-7 bind to less conserved RBD epitopes, which do not overlap with the epitope of aRBD-2, and do not show significant binding with RBD of certain variants [27]. In this approach, the immune diversity of alpacas simultaneously immunized with the SARS-CoV-2 S protein trimer and a DNA vaccine (encoding the SARS-CoV-2 S protein ectodomain) was leveraged to obtain a phage library with a titre of 8 × 10^13^ CFU/mL. Nanobody selection was performed using three rounds of biopanning with the Wuhan-Hu-1 RBD protein (GenBank: MN908947). Based on ELISA and pseudovirus neutralization results, three nanobody variants were selected, capable of competitively inhibiting ACE2 (angiotensin-converting enzyme 2) receptor binding with RBD and neutralizing pseudoviruses of SARS-CoV-2 variants D614G, Alpha, Beta, Gamma, and Delta, and Omicron sublineages BA.1, BA.2, BA.4, and BA.5. One nanobody (aVHH-13-Fc) effectively protected hamsters from infection by the SARS-CoV-2 prototype, Delta, and Omicron BA.1 and BA.2. Structural modeling of the aVHH-13 and RBD complex showed the possibility of the nanobody binding to the receptor-binding motif of RBD [28]. In study [9], an even more complex strategy was implemented. The alpaca was immunized six times: three times with recombinant RBD, once with viral particles of the AdC68-19S vaccine expressing the S trimer, and twice with recombinant S-2P protein (Wuhan-Hu-1). The yeast library was enriched through one round of MACS biopanning (magnetic-activated cell sorting), followed by an additional FACS biopanning round (fluorescence-activated cell sorting). In both cases, the SARS-CoV-2 S-2P protein was used. Three nanobodies (1-2C7, Nb70, 3-2A2-4), which exhibited varying degrees of competition with ACE2 and eCR3022 (an antibody targeting a hidden epitope on the RBD, accessible only when the RBD is in the “up” conformation [29]), were analyzed for neutralizing activity against different SARS-CoV-2 variants. This comprehensive approach enabled the selection of a group of nanobodies with efficacy against SARS-CoV-2 variants Alpha, Beta, Gamma, and Delta, and Omicron subvariants BA.1, BA.2, and BA.4/5, as well as SARS-CoV-1.

These studies highlight that the sequence of biopanning and the choice of antigens play a crucial role in selecting broadly neutralizing nanobodies. Expanding on this concept, our study involved four rounds of biopanning using a phage display nanobody library derived from an alpaca immunized with the RBD of the Wuhan-Hu-1 variant [30]. We also explore the feasibility of conducting more than three rounds of biopanning with sequential use of different variants of SARS-CoV-2 surface antigens. The goal of our study was to identify nanobodies capable of neutralizing a broad spectrum of SARS-CoV-2 variants.

## 2. Materials and Methods

### 2.1. Recombinant Antigens

Recombinant RBD and S protein trimers of SARS-CoV-2 variants Wuhan Hu-1, Beta (B.1.351), Delta (B.1.617.2), and Omicron (B.1.1.529), obtained from the Bioengineering Department of the State Research Center of Virology and Biotechnology “Vector” Rospotrebnadzor (Federal Service for Surveillance on Consumer Rights Protection and Human Wellbeing), were used in the study. These variants will hereinafter be referred to as Wuhan, Beta, Delta, and Omicron.

### 2.2. Phage Display Biopanning Against SARS-CoV-2

The only difference in each round of biopanning was the target with which the phage clones interacted. Recombinant RBD variants of Wuhan Hu-1, Beta (B.1.351), and Delta (B.1.617.2) were used as targets in the first three rounds. In the fourth round, a recombinant S protein trimer of the Omicron variant (B.1.1.529) was used. Recombinant protein (150 ng per well) was adsorbed onto 96-well plates in a coating buffer (0.1 M NaHCO_3_) and incubated for 16 h at 4 °C. After incubation, blocking was performed by adding 150 µL of blocking buffer (0.1 M NaHCO_3_ with 0.5% *v*/*v* BSA) and incubating for 1 h at 4 °C. The plate was washed with TBS-T buffer (tris-buffered saline with 0.1% *v*/*v* Tween-20). After washing, a phage suspension with a titre of 1 × 10^10^ CFU/mL (100 µL) was added and incubated for 1 h at room temperature. Following incubation, the wells were washed again with wash buffer, and 100 µL of elution buffer (0.2 M Glycine-HCl) was added to each well and incubated at room temperature for 15 min on a shaker set at 25 rpm. Then, 15 µL of neutralization buffer (1 M Tris-HCl) was added to stop the reaction. The eluates obtained after biopanning were amplified and the nucleotide sequences of individual phage clones were sequenced.

### 2.3. Solid-Phase Enzyme-Linked Immunosorbent Assay of Amplified Phage Clones

Recombinant trimers of three SARS-CoV-2 variants (Wuhan Hu-1, Delta (B.1.617.2), and Omicron (B.1.1.529)), dissolved in 0.1 M NaHCO_3_, were added at 200 ng per well and incubated for 16 h at 4 °C. After incubation, the solution was removed and 200 µL of blocking buffer (0.1 M NaHCO_3_ with 0.5% *v*/*v* BSA) was added, followed by incubation for 2 h at 37 °C. Washing was performed three times using TBS-T wash buffer (tris-buffered saline with 0.5% *v*/*v* Tween-20). After washing, 150 µL of amplified phage clones, previously diluted in the blocking solution to a titre of 10^9^ CFU/mL, were added to the wells. Human polyclonal serum from a recovered COVID-19 patient and phage helper M13K07 were used as positive and negative controls, respectively. After incubation at 37 °C for 2.5 h, a sevenfold wash was performed, and conjugates were added, HRP Anti-M13 Bacteriophage (Abcam, Cambridge, UK) at a dilution of 1:4000 and anti-Human IgG-HRP (Thermo Fisher Scientific, Waltham, MA, USA) at a dilution of 1:5000, followed by incubation for 1.5 h at 37 °C. After incubation, the plate was washed again. To detect the signal, 100 µL of TMB was added to each well, and the reaction was stopped by adding 100 µL of 1 M HCl. Optical density (OD) was measured using a Thermo Scientific Varioskan LUX multimode reader at a wavelength of 450 nm.

### 2.4. Cloning of Nanobody Genes into Expression Vector

For the amplification of nanobody genes, primers Phage-F (5′-aaaaaaCATATGAAATACCTATTGCCTACGGCA-3′) and Phage-R (5′-aaaaaaGTCGACACCACTACCGCTACCTGAGGAGACGGTGACCTGGG-3′) were designed, incorporating sequences corresponding to the FauNDI and SalI restriction sites for subsequent cloning of the gene into the pET21 vector. Phagemids from individual phage clones were used as templates for nanobody gene amplification.

The analysis of amplicons was performed by electrophoretic separation of the reaction mixture in a 1% agarose gel. The cleavage products were purified from the agarose gel and ligated using T4 bacteriophage ligase. Clones containing the insert were verified by Sanger sequencing for the presence of deletions and substitutions.

### 2.5. Production of Nanobodies

For the production of recombinant nanobody-expressing strains, *E. coli* BL21(DE3) strain (Novagen, Madison, WI, USA) was used. The cells transformed with the vector containing the insert were selectively cultured in 100 mL of liquid LB medium supplemented with sodium ampicillin at a working concentration of 20 µg/mL. The synthesis of the target recombinant protein was induced by 0.5 mM IPTG (isopropyl-β-D-thiogalactoside) and vigorous shaking (180 rpm) in a thermoshaker at 30 °C. Clones of nanobody-producing strains were selected based on the presence of the target protein, determined by electrophoresis of cell lysates in 10% SDS-PAGE (sodium dodecyl sulfate-polyacrylamide gel electrophoresis). As a control, induced cell lysates from *E. coli* BL21(DE3) strain, containing the empty pET21(-) vector, were used.

### 2.6. Competitive Enzyme-Linked Immunosorbent Assay (ELISA)

The specificity of the synthesized nanobodies was assessed by competitive ELISA using recombinant trimeric S proteins from SARS-CoV-2 variants—Wuhan, Delta, and Omicron. Recombinant S protein trimers of SARS-CoV-2, adsorbed for 16 h at 4 °C at 400 ng per well, were incubated for 1.5 h with lysates of nanobody producers for four variants. Human broadly neutralizing monoclonal antibody against SARS-CoV-2, iB20, obtained at the Institute of Molecular and Cellular Biology, SB RAS [31,32], was used as a positive control. The negative control consisted of the lysate of induced cells transformed with the empty pET21 plasmid, and the heterologous nanobody control was VHH9, a nanobody specific for HIV (previously described in [33]), obtained at the Immunochemistry Laboratory of the State Research Center of Virology and Biotechnology “Vector”, Rospotrebnadzor. Then, recombinant ACE2 (diluted 1:1000) and secondary Anti-Human IgG-HRP antibodies (diluted 1:5000) were added simultaneously, followed by incubation for 1.5 h. The sequential addition of lysates and conjugates was necessary because ACE2 was conjugated with HRP. The signal level of direct interaction between the recombinant trimer and ACE2 was considered as 100% interaction. A positive result was considered when the OD at 450 nm in the sample was lower than in the negative control.

### 2.7. Nanobody Purification

Recombinant proteins containing a polyhistidine tag were purified using affinity chromatography on Ni-chelate resin IMAC Sepharose 6 Fast Flow (GE HealthCare, Uppsala, Sweden). A solution containing the protein was applied to the column pre-saturated with 0.2 M NiCl2 resin and sequentially washed with two washing buffers (tris-glycine buffer containing 10 mM and 100 mM imidazole). Target proteins were eluted using tris-glycine buffer containing 300 mM imidazole. The purified recombinant proteins were analyzed by SDS-PAGE protein electrophoresis according to the Laemmli method. The concentration of recombinant proteins was determined by measuring the absorbance at 280 nm using a NanoDrop OneC spectrophotometer (Thermo Fisher Scientific, Waltham, MA, USA).

### 2.8. Virus Neutralization Assay

The titre of virus-neutralizing nanobodies was determined using a virus neutralization assay. In this study, the following SARS-CoV-2 strains were used: Wuhan (hCoV-19/Australia/VIC01/2020), Delta (hCoV-19/Russia/PSK-2804/2021), Omicron 1 (hCoV-19/Russia/Moscow171619-031221/2021), and XBB.1.5 (hCoV-19/Russia/TYU-SRC-8642/2023). They were obtained from the State Collection of Pathogens of Viral Infections and Rickettsioses, State Research Center of Virology and Biotechnology “Vector”, Rospotrebnadzor. Monoclonal nanobody preparations (at a concentration of 1 mg/mL) were mixed with the virus in different dilutions in a 1:1 ratio and applied in duplicates to a monolayer of Vero cell cultures. Any specific cell culture damage in the well was recorded as CPE (cytopathic effect, structural changes in host cells caused by viral infection). The titre was defined as the last dilution at which protection of the monolayer of cell culture in the well from the viral CPE was observed. As a positive control, a 20-fold dilution of serum from a COVID-19 convalescent patient with a previously established titre of 1:80 was used. The negative control consisted of the growth medium.

### 2.9. Statistical Analysis

Statistical analysis of the obtained data was performed using Excel 2019 (Microsoft Corporation, Redmond, WA, USA). For quantitative variables, the results are presented as the mean (M) with the standard deviation (±SD). The data were obtained from three replicates within a single experiment (*n* = 3). The standard deviation did not exceed 3.5% of the mean.

To assess statistical differences between nanobodies, one-way ANOVA (single factor) was performed separately for each SARS-CoV-2 variant (Wuhan, Delta, and Omicron). The significance threshold was set at *p* < 0.05.

### 2.10. Molecular Modeling

The tertiary structure of the KWL nanobody was obtained using the AlphaFold methodology [34]. The AlphaFold2 software (v2.3.2, monomer prediction mode, using a full database dated no later than 18 November 2023) was installed on the SKIF Shared Use Center cluster. The launch parameters were monomer mode using the complete database, dated no later than 18 November 2023. Optimal protein structures were selected based on their respective scoring function values. Geometric parameters were refined using molecular dynamics simulations of the protein over 200 ns.

The geometric parameters of the receptor-binding domain of the S protein for three variants were downloaded from the non-commercial Protein Data Bank [35]: Wuhan (PDB ID: 7WNB), Delta (B.1.617.2) (PDB ID: 8I5H), and Omicron (BA.1) (PDB ID: 7YOW) [36]. The proteins were checked for errors using the Schrodinger Protein PrepWizard plugin. If necessary, missing amino acid side chains were added, bond multiplicities were restored, solvent molecules were removed (when needed), and hydrogen atoms were added and minimized. When preparing the surface protein calculations, the pH of the environment was considered, and the geometric parameters were optimized using the OPLS4 force field method (a molecular mechanics model for calculating interatomic interactions and optimizing molecular geometry) [37].

To assess the potential interaction of the nanobody–antigen complexes, the protein–protein docking protocol PIPER was used [38]. After thorough analysis, the optimal docking position of the nanobody–antigen complex was selected for subsequent refinement of the geometric parameters using molecular dynamics simulations.

The nanobody–antigen complex was placed in a cubic box with a 20 Å buffer. The box was filled with a 0.15 M NaCl aqueous solution. The TIP3P water model (Transferable Intermolecular Potential 3-Point, a simplified representation of water molecules in molecular dynamics simulations) was used. Counterions (Na⁺ and/or Cl⁻) were added to neutralize the system’s charge. All models were initially subjected to a relaxation procedure for 2 ns to minimize internal stresses. The RESPA integrator algorithm (a numerical method for efficient long-timescale molecular dynamics simulations) was used with a 2 fs time step. Temperature control was maintained using the Nosé–Hoover thermostat (a method for regulating temperature in molecular dynamics simulations), and the NPT thermodynamic ensemble was applied. The simulation time was set to 200 ns, with 10,000 frames analyzed. Open-source Desmond software (v.7.2) was used [39]. The MD simulation trajectory analysis and visualization of the calculations were performed using VMD (visual molecular dynamics) [40]. Additionally, clustering of nanobody–antigen complex structures was performed based on RMSD (root mean square deviation, a measure of structural deviation over time) for the last 100 ns of the MD simulation to find the statistically most likely protein orientation in space.

## 3. Results

From the phage display nanobody library (biological titre 3 × 10^10^ CFU/mL), generated based on the immune repertoire of a llama immunized with the Wuhan RBD, 30 phage clones were selected after three rounds of biopanning. Recombinant RBD of the SARS-CoV-2 spike protein variants Wuhan Hu-1, Beta (B.1.351), and Delta (B.1.617.2) were sequentially used as targets in the three rounds of selection. An additional fourth round was performed using the recombinant S protein trimer of the Omicron (B.1.1.529) variant, and 24 additional phage clones were selected. In total, 54 phage clones were selected from the phage repertoire in the library (Figure 1).

The selected phage clones were checked by sequencing for mutations and deletions. Sequencing data of the nucleotide sequences encoding the nanobody showed that correctly assembled genes without deletions, frame shifts, and stop codons were found only in 22 nanobody gene variants (15 after the third round and 7 after the fourth round of selection).

Phage clones with functional genes were amplified and tested by ELISA for binding to recombinant S protein trimers from three SARS-CoV-2 variants—Wuhan, Delta, and Omicron. The selection of trimers was based on the desire to assess the recognition ability of the RBD by the selected phage clones in the context of the full-length S protein. Fifteen phage clones showed varying degrees of interaction in the ELISA (presented in the Appendix A, Appendix A). On average, the OD of phage clones selected after the third round was higher for the Delta variant (OD > 3–4) compared to the Wuhan and Omicron trimers (OD > 2–3). One phage clone showed no reactivity. As a result of the ELISA performed with phage clones selected after three rounds, 14 out of 15 nanobody-expressing phage clones were selected for further work, as they showed reactivity with at least one of the three trimer variants. After the fourth round of biopanning, seven nanobodies were selected for analysis in ELISA, as the genes of the others contained stop codons. Of the seven selected phage clones expressing unique nanobodies, only one specifically interacted with all three variants of the S protein trimer. However, the interaction signal with the Omicron trimer (OD 2.8) was lower than with the Wuhan and Delta trimers (OD > 4.4). Thus, after four rounds of biopanning, seven phage clones showed no reactivity and likely contain variants of non-specific nanobodies that were randomly selected during the selection process.

Nucleotide sequences encoding the nanobodies that showed specific interactions in ELISA with recombinant trimers were cloned into the pET21 vector. The resulting recombinant plasmids were used to transform *E. coli* cells (BL21(DE3) strain). For four of the thirteen nanobody-producing strains, after optimizing expression conditions, nanobody accumulation (in soluble form) was detected, with a yield of ≥30% of total cellular protein, which was sufficient for immunochemical analysis (Figure 2).

The obtained nanobodies were tested for interaction with recombinant surface S protein in a competitive ELISA. Sequential addition of recombinant nanobodies and ACE2 to immobilized antigens allows the assessment of nanobody interaction with the target protein in the presence of competing ACE2. The results of ACE2 binding to recombinant S protein trimers of SARS-CoV-2, in the presence of inhibition by nanobodies, are shown in Figure 3.

Three nanobodies (PRV, KWL, and SKP) inhibited the interaction between ACE2 and recombinant S protein trimers of the SARS-CoV-2 Wuhan variant and at least one other variant. Two of them (KWL and SKP) inhibited binding to the Delta variant but not to Omicron. The PRV nanobody, which blocked the interaction between ACE2 and the Wuhan trimer, also inhibited binding to Omicron but not to Delta. The inhibition values for the RC nanobody differed from the negative control by no more than 20%.

The results of virus neutralization for SARS-CoV-2 variants Wuhan, Delta, Omicron, and the subvariant XBB suggest that virus-neutralizing activity is present in all four nanobodies, though to varying degrees (Table 1). The virus neutralization titre for all nanobodies was highest for the Delta variant. For the PRV nanobody, the neutralization titres for the Wuhan and Delta variants were the same, but the titre for Omicron was higher. The KWL nanobody neutralized both the Wuhan and Omicron variants at the same titres. However, the titre for the Delta variant reached 1/2560. In contrast to the other nanobodies, the SKP nanobody exhibited the highest neutralizing activity against the XBB subvariant (titre 1/20).

To construct a three-dimensional model of the interaction between the obtained nanobodies and the RBD of the SARS-CoV-2 S protein, a number of theoretical approaches were used. The KWL nanobody, which exhibited the highest neutralizing activity (titre 1/2560) among the obtained nanobodies against the SARS-CoV-2 Delta variant, was selected for the study. The analysis of the KWL nanobody folding procedure, followed by the refinement of its tertiary structure using molecular mechanics methods, is presented in the Appendix A (Appendix A). The prediction result using the AlphaFold methodology can be considered successful. For most of the protein fragments, the internal quality assessment score, pLDDT (predicted local distance difference test), exceeds 90. As a result of the molecular (protein–protein) docking procedure, 30 docking solutions were obtained. For each complex, positions were selected (Appendix A) where the nanobody is located at the interface of the receptor-binding domain (RBD) with ACE2. The selected positions are characterized by the highest number of intermolecular contacts (such as hydrogen bonds, salt bridges, and π-π stacking interactions) with a relatively small number of undesirable clash interactions. These positions were used for subsequent molecular dynamics simulations.

The graphs showing the RMSD of the atom positions in the nanobody–antigen complexes, derived from the analysis of the molecular dynamics simulation trajectory, indicate that by 100 ns, the positions of the amino acid residues of nanobody (KWL) and RBD from different variants converged (Appendix A). The clustering procedure allowed the identification of statistically significant nanobody–RBD complexes that were most frequently realized during the last 100 ns of the simulation. The KWL nanobody binds to the interface of the RBD–ACE2 contact area (Figure 4a). In the case of the interaction between KWL and RBD of the Wuhan and Delta variants, the nanobody almost completely covers the interaction area (Figure 4b,c). The CDR loops’ amino acid residues of the nanobody form intermolecular interactions with the amino acids of the so-called three contact zones of the receptor [41,42]. Namely, hydrogen bonds were observed between the following pairs of amino acids of the nanobody and the antigen in the KWL–RBD–Wuhan complex: D62–Q498, Y108–G502, N107–Y505, W111–S494, Y114–S494, D112–Y453, Y95–K484 (Figure 4b). In the KWL–RBD–Delta complex, they were observed between the following pairs: Y95–T500, R45–N501, R45–Y449, G118–Y505, W117–Y505; E113–S494; Y104–Y489 (Figure 4c). Additionally, salt bridges were observed between the pairs of amino acids R45 and G113, and R45 and K417 (in the KWL–RBD–Wuhan complex), and between D112 and K417 (in the KWL–RBD–Delta complex), and π-π stacking interactions were observed between the aromatic rings Y108–Y505 (KWL–RBD–Wuhan complex) and Y116–Y499 (KWL–RBD–Delta complex) (Figure 4b,c). In works [41,42], the amino acids of the N501 and E484 domains are identified as key amino acids that form intermolecular contacts with the amino acids of ACE2.

The positioning of the KWL nanobody relative to the Omicron RBD in the KWL–RBD–Omicron complex differs significantly from the other complexes. In fact, the amino acid residues of the CDR loops of the nanobody interact with only one of the three contact zones of the RBD–ACE2 complex (Figure 4d). Hydrogen bonds are observed between the following pairs of amino acids: R45–N487, W117–Y489, Y116–N477, and E113–K458. A salt bridge was observed between E113 and K458, and π–π stacking interactions between the aromatic rings Y103 and F456. The positioning of the nanobody relative to the antigen affects the value of the binding free energy (ΔG_bind_), which is a measure of the thermodynamic stability of molecular interactions (Figure 4). The KWL–RBD–Wuhan complex was found to be the most thermodynamically stable, with the lowest binding free energy. The KWL–RBD–Delta complex is considered slightly less stable. The binding of the nanobody to the receptor-binding domain of the Omicron variant is characterized by the highest binding energy, indicating a less stable (from a thermodynamic perspective) state of the system.

## 4. Discussion

For the successful selection of target nanobodies using phage display technology, two conditions must be met: the use of an appropriate phage nanobody library and a properly selected biopanning strategy [43]. An analysis of studies on nanobody discovery against SARS-CoV-2 shows that researchers aiming to increase the breadth of neutralization typically attempt to use phage libraries based on the immune repertoire of animals immunized sequentially with multiple SARS-CoV-2 variants [44]. However, this approach results in the identification of new nanobody variants lagging behind the virus’ evolution speed, as the creation of a new library takes time. An alternative approach may involve searching for nanobody variants with broad reactivity in the original library (derived from the immune repertoire of animals immunized only with the Wuhan variant). For biopanning, recombinant RBD or S protein repertoires from various SARS-CoV-2 variants (Beta, Delta, Omicron, etc.) can be used. In our study, 54 phage clones were selected from the phage nanobody library after four rounds of biopanning (30 after the third round and 24 after the fourth round). The selection of this strategy (Wuhan → Beta → Delta → Omicron) was based on the hypothesis that sequential use of heterologous antigens in the order of their natural emergence reproduces the viral evolutionary process in vitro, facilitating the selection of broadly neutralizing nanobodies. This approach may help preserve nanobody variants specific not only to Omicron but also to earlier variants such as Wuhan and Delta. The selection of individual clones after the third and fourth rounds was aimed at evaluating the impact of the fourth round on the enrichment of the phage library and the characteristics of the selected nanobodies. Sequencing results showed the presence of stop codons or frame shifts in the nanobody sequences of 32 out of 54 phage clones. The sequencing results revealed the presence of amber stop codons in the nanobody sequences of 32 out of 54 phage clones. As these clones were unable to produce full-length VHH under standard expression conditions, they were excluded from further analysis. The proportion of phage clones carrying defective genes was 21% higher among those selected after the fourth round. This likely indicates the accumulation of clones with defects in the nucleotide sequence with each subsequent round of biopanning [43].

In the study [26], a comparison of phage clones selected after the first and second rounds showed that 61% and 97%, respectively, specifically interacted with RBD. The analysis of the binding of 22 phage clones with functional genes, selected in our study, to three variants of SARS-CoV-2 S protein trimers (Wuhan, Delta, and Omicron) showed that 32% of them did not bind to the recombinant trimers. Most of the non-specific clones were selected after the fourth round. This raises the question of the feasibility of conducting a fourth round of biopanning. At the same time, conducting only one or two rounds of biopanning may significantly limit the enrichment of the library [43]. In practice, this means that for the selection of highly specific clones, it is sufficient to perform no more than two or three rounds of biopanning.

ELISA with phage clones selected through biopanning not only allowed the assessment of the specificity of individual phage nanobodies but also enabled further selection against random variants. Of the 22 phage clones with functional genes, only 68% showed specific interaction in ELISA with recombinant trimers of the Wuhan, Delta, and Omicron S proteins. The Beta variant was not included in ELISA as antigens were selected based on their use in viral neutralization assays, and this variant was not available to our research group. In studies using the antigen of only the Wuhan variant, the proportion of specific clones was higher [26,45]. It seems that when biopanning is performed using antigens of only one variant (e.g., Wuhan), the phage library is enriched with clones carrying nanobodies with similar paratopes (antigen-binding sites of the nanobody). Probably, when additional heterologous variants (i.e., Delta, Omicron) are used, each subsequent round of biopanning leads to the loss of some nanobody variants specific to the antigen from the previous round. It is also interesting to note that testing the neutralizing activity of the nanobodies selected in our study showed that they neutralized the Delta variant the most, despite the fact that the library was generated based on diversity after immunization with the Wuhan variant. The recombinant RBD of the Delta variant was used for the third round of biopanning, which likely led to a shift in the diversity of the eluate and the subsequent variants of the nanobodies obtained. This effect is not uncommon. At each new stage of biopanning, the phage population displaying nanobodies interacts with a newly introduced antigen. The selective pressure imposed by these antigens can shift the enrichment process toward clones with the highest affinity for the current target, potentially leading to the loss of lower-affinity clones that may possess broader neutralizing potential. To mitigate this shift in diversity, an alternative approach could involve performing several initial rounds of selection on a single antigen (e.g., Wuhan) before introducing a mixture of antigens (e.g., Wuhan + Delta + Omicron) in later rounds. This strategy may help maintain a broader repertoire of nanobody variants and prevent the complete displacement of earlier selected clones, thereby enhancing the likelihood of isolating broadly neutralizing nanobodies.

It seems that the presence of phage clones expressing non-specific nanobodies on their surface in the eluates, even after several rounds of biopanning, is a common occurrence [26,28,45]. This indicates the inevitable loss of nanobody variants. In our study, we also observed that each standard step of phage display led to the loss of nanobody variants (Figure 5). This highlights the need to select and analyze the maximum possible diversity of phage clones after biopanning. In this regard, we believe that a limitation of our study is the number of phage clones selected for analysis after the third and fourth rounds.

Despite the limitations of our study related to the small number of selected phage clones, we identified variants that exhibit not only cross-reactivity with a panel of S proteins but also the ability to neutralize strains from different SARS-CoV-2 variants. One of the nanobodies demonstrated a neutralizing activity of 1:2560 against the Delta variant, which is comparable to the iB20 antibody (unpublished data) [32]. For this nanobody variant, we attempted to construct a structural model of its interaction with the RBD to partially explain the observed results. It is known that the RBD of the S protein is a region with high amino acid variability [36]. The sequence identity between the RBDs of the Wuhan and Delta (B.1.617.2) variants is 99%, while a comparison between Wuhan and Omicron (BA.1) reveals 92% sequence identity. Most of this variability is concentrated in the first RBD epitope, which significantly overlaps with the ACE2 binding interface. According to our model, the KWL nanobody binds to the contact region of the RBD–ACE2 interface (Figure 4a). In the case of KWL interaction with the RBDs of the Wuhan and Delta variants, the nanobody almost completely covers the binding region (Figure 4b,c). However, the spatial positioning of the nanobody relative to the RBD of the Omicron variant in the KWL–RBD–Omicron complex differs significantly from that of other complexes. The results of molecular modeling are generally consistent with experimental data: The KWL–RBD–Omicron complex exhibits the highest binding energy (Figure 4), while the KWL nanobody is less effective at inhibiting ACE2 interaction with the recombinant S protein trimer of the Omicron variant (Figure 3). Meanwhile, the binding energies of the KWL–RBD–Wuhan and KWL–RBD–Delta complexes are relatively similar, which aligns with their ability to inhibit ACE2 binding to the S protein trimers of these variants. However, these results partially contradict the neutralization data. Despite similar binding energies, the neutralization titer of KWL against the Delta variant was significantly higher (1:2560) than it was against Wuhan (1:80). One possible explanation for this discrepancy is the difference in the spatial positioning of KWL, which, in the case of Delta, may impose greater steric hindrance on the interaction between the RBD and ACE2. Additionally, mutations characteristic of Delta (e.g., L452R and T478K) may alter the conformational flexibility of the protein, and KWL may stabilize the RBD in a less active state, hindering its binding to the receptor. Other neutralization mechanisms cannot be ruled out, including effects on the dynamics of the trimeric S protein or its stabilization in a closed (“down”) conformation.

Similar discrepancies were also observed in the study of other nanobodies. For example, the PRV nanobody neutralizes both the Wuhan and Delta variants with identical titers (1:640) but does not inhibit ACE2 binding to the S protein trimer of the Delta variant. A similar inconsistency is observed for KWL, which inhibits binding to Wuhan but not to Omicron, while its neutralization titers for these variants are identical (1:80). This is most likely due to the fact that blocking the RBD–ACE2 interaction is unlikely to be the sole mechanism of neutralization. Some nanobodies may stabilize the S protein in an inactive state, preventing its transition to a conformation required for receptor binding. In this case, they may not block ACE2–RBD interaction in a static ELISA assay but may still influence the infection process under viral neutralization conditions. A similar case was described in study [9], where the nanobody 3-2A2-4 exhibited broad and potent neutralization, including against the Omicron variant, despite weak competition with ACE2, supporting the existence of alternative viral neutralization mechanisms.

Thus, the biopanning strategy we developed, using various RBD variants, allowed the selection of nanobodies that neutralize a range of SARS-CoV-2 variants. At the same time, we observed a tendency for the reactivity of the selected variants to shift towards the Delta variant. Each step in the process of nanobody production, from analyzing the reactivity of phage clones to generating bacterial expression systems, is associated with the loss of variants. Therefore, a successful search should involve the use of the maximum possible number of nanobody variants. Future studies may focus on a detailed characterization of nanobody interactions, including epitope binning and binding kinetics analysis, to gain a deeper understanding of their specificity and affinity. These studies will help clarify the mechanisms of neutralization and may contribute to assessing the potential of the selected nanobodies.

## Figures and Tables

**Figure 1 antibodies-14-00023-f001:**
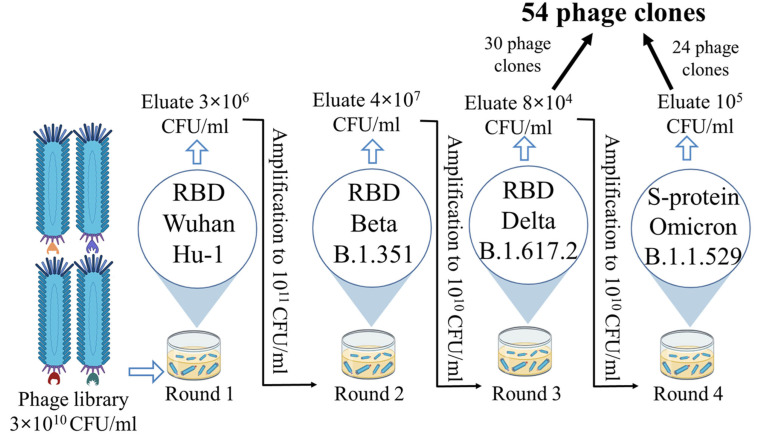
Phage display scheme showing antigens, eluate, and amplicon titres for each round.

**Figure 2 antibodies-14-00023-f002:**
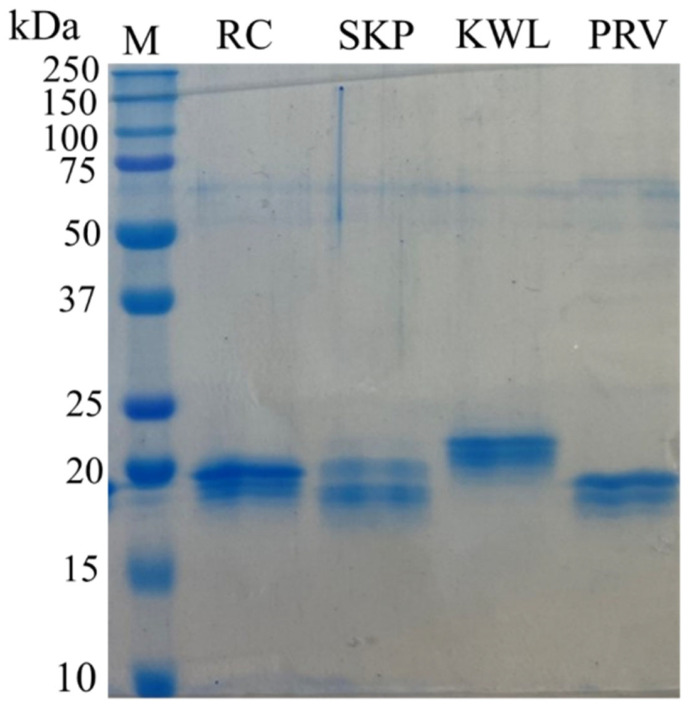
Electrophoretic separation of synthesized nanobodies in 10% SDS-PAGE. Labels: M—molecular weight protein markers with the molecular weight in kDa indicated on the left (Precision Plus Protein™ Dual Xtra Prestained Protein Standards, Bio-Rad, Hercules, CA, USA); RC, SKP, KWL, PRV—purified recombinant nanobodies.

**Figure 3 antibodies-14-00023-f003:**
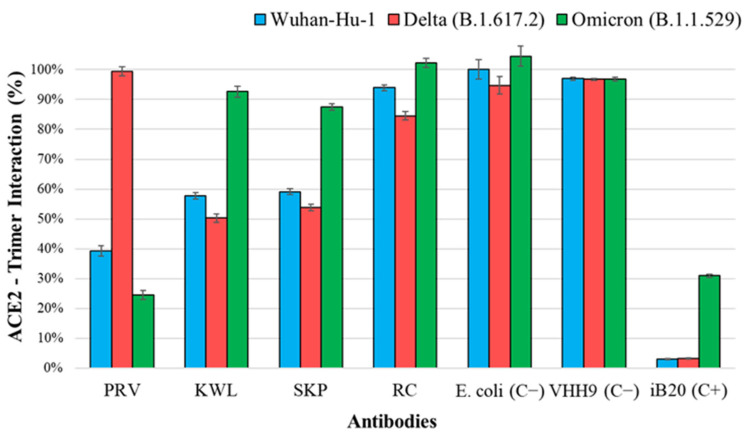
Binding of ACE2 to recombinant SARS-CoV-2 S protein trimers upon inhibition of interaction by nanobodies. The 100% interaction level is considered to be the signal of direct binding between the trimer and ACE2. Notations: PRV, KWL, SKP, RC—lysates of nanobody producers; *E. coli* (C−)—negative control producer, lysate of cells transformed with the pET21a(−) plasmid; VHH9 (C−)—nanobody specific to HIV-1, negative control of a heterologous nanobody; iB20—broad-neutralizing human monoclonal antibody against SARS-CoV-2 [32], positive control. One-way ANOVA showed statistically significant differences in the inhibition of ACE2 binding among nanobodies for each SARS-CoV-2 variant (Wuhan, Delta, and Omicron) with *p* < 0.0001.

**Figure 4 antibodies-14-00023-f004:**
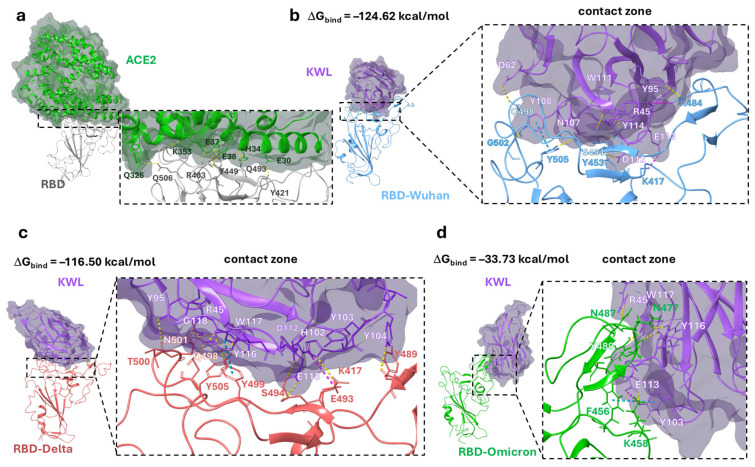
Position of the KWL nanobody in the ACE2-binding domain. (**a**)—visualization of the ACE2–RBD complex (PDB ID 6VW1 [42]); (**b**)—statistically significant KWL–RBD–Wuhan complex; (**c**)—statistically significant KWL–RBD–Delta complex; (**d**)—statistically significant KWL–RBD–Omicron complex, obtained as a result of clustering the last 100 ns of MD simulation. For better visual perception, the structure of each protein, including α-helices and β-strands, is shown in different colors. In panel (**a**), ACE2 is shown in green, in panels (**b**–**d**), the KWL nanobody is shown in purple, RBD Wuhan in blue, RBD Delta in red, and RBD Omicron in green. Hydrogen bonds, salt bridges, and π-π stacking interactions are shown as yellow, purple, and blue dashed lines, respectively.

**Figure 5 antibodies-14-00023-f005:**
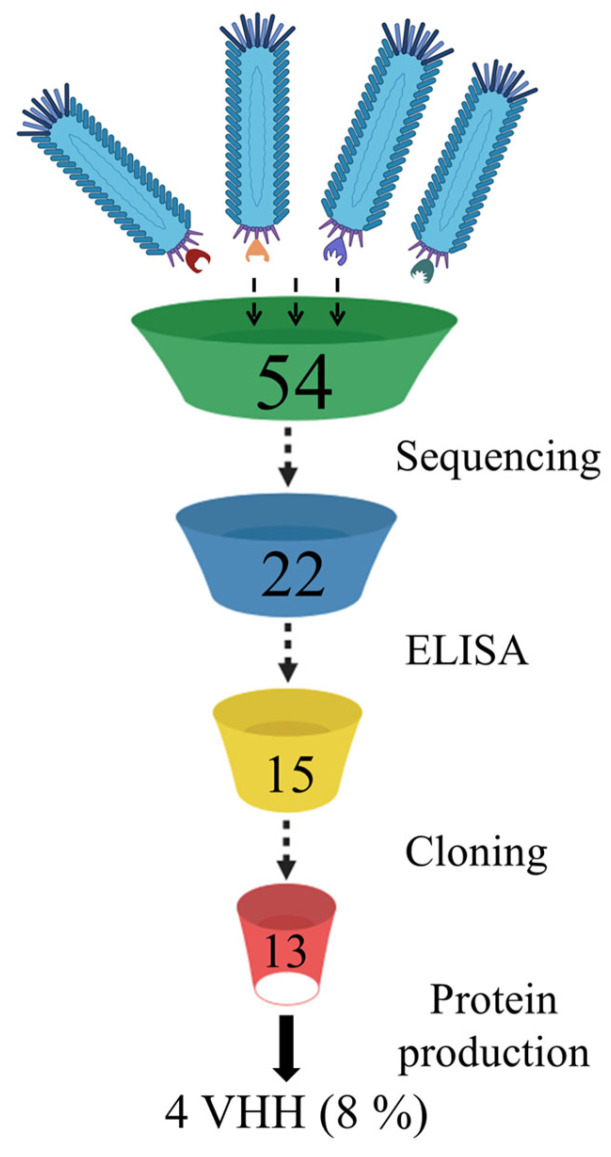
Loss of nanobody variant diversity during standard phage display procedures.

**Table 1 antibodies-14-00023-t001:** Neutralization titre of four virus-neutralizing nanobodies against four SARS-CoV-2 variants. To enhance the visual perception of numerical values, a color scheme has been applied in the table. Lighter colors represent lower values, while higher titers are displayed in progressively darker shades.

	Wuhan	Delta	Omicron 1	XBB 1.5
PRV	640	640	1280	<10
KWL	80	2560	80	10
SKP	40	640	80	20
RC	40	640	40	<10

## Data Availability

The sequences of the nanobodies obtained in this study are available upon request.

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
