# Peer review of "The Use of Heterologous Antigens for Biopanning Enables the Selection of Broadly Neutralizing Nanobodies Against SARS-CoV-2"

_2073-4468, 2025, doi:10.3390/antib14010023_

Round 1

Reviewer 1 Report

Comments and Suggestions for Authors

The authors present a study detailing the use of biopanning to select nanobodies, with broad binding capabilities against SARS-CoV-2 variants. Whilst the reviewer appreciates the effort put into this study, they suggest several key shortcomings which would reduce the suitability for publication in this current state. 

  • Of major issue is the set order of panning. Would the authors not agree that examining clones generated following panning the respective variants in a different order would greatly improve the quality of the manuscript and the relevancy of the data presented? Much more rationale needs included for the panning strategy and order used.
  • Additionally, it was claimed "Three nanobodies (PRV, KWL, SKP) were able to inhibit the interaction between 
    ACE2 and recombinant S protein trimers of SARS-CoV-2 variants Wuhan, Delta, and Omicron". This does not appear to be the case. Maybe this is a typographical error?
  • Further statistical analysis should be included throughout e.g. Fig 3
  • Consistent nomenclature - nanobodies, antibodies, should be used throughout
  • Further evaluation of the binders e.g. binding kinetics, epitope binning etc should be included
  • Why were the phage clones not tested by ELISA against RBD Beta?

The reviewer would appreciate a consideration of these points and looks forward to a response

Reviewer 2 Report

Comments and Suggestions for Authors

The manuscript (Antibodies 3477949) by Aripov et al. reports a study that employs a heterologous antigen biopanning strategy to develop broadly neutralizing antibodies against evolving SARS-CoV-2 variants. The authors successfully isolated four nanobodies targeting the receptor-binding domain (RBD) of the SARS-CoV-2 spike protein, demonstrating cross-neutralization against a broad spectrum of COVID variants. This manuscript provides valuable insights into nanobody development against SARS-CoV-2 variants, and the use of heterologous biopanning strategy represents a solid advance in this field. I believe that this manuscript is suitable for publication, provided that the following concerns are properly addressed.

Major concerns:

  1. The use of a heterologous biopanning strategy to develop nanobodies against multiple COVID variants is the key innovation of this work. Please provide more explanation regarding the rationale behind your strategy design. For example, why are the antigens in rounds 1 through 4 arranged in the order of Wuhan, Beta, Delta, and Omicron variants? Shouldn’t the process start with the most prevalent variants and then evaluate whether the resulting candidates can also target the other variants?
  2. In total, 54 phage clones were selected from the library in rounds 3 and 4, but only 22 of them exhibited correctly assembled genes. First, the authors should discuss whether this sample size sufficiently captures the library’s diversity. Second, the authors should address the potential reasons for the low success rate (22/55). Including sequencing information for both functional and non-functional clones may be helpful.

Minor concerns:

  1. Figure 3 and Table 1 would benefit from the inclusion of error bars or statistical significance indicators to better demonstrate the repeatability of these experiments.
  2. Using simpler language or providing brief explanations to clarify very specialized terms (e.g., paratopes, OPLS4 force field, etc.) would enhance readability.
  3. Please ensure that all abbreviations are defined upon their first use. For example, the abbreviations “CFU”, “MACS”, and “FACS” are used without prior definition.
  4. While KWL’s binding energy for the Wuhan RBD is low (indicating stability), its neutralization titer for the Wuhan variant is unexpectedly high. Please discuss in more detail the discrepancy between the modeling results and the neutralization data, as well as the potential underlying mechanisms. The authors provided an example stating, “The nanobody 3-2A2-4, with high neutralization values, had minimal weak competition with ACE2.” However, this represents a completely different scenario, which cannot be used to justify the case of KWL.

Reviewer 3 Report

Comments and Suggestions for Authors

In this study titled "The Use of Heterologous Antigens for Bio panning Enables the Selection of Broadly Neutralizing Nanobodies Against SARS-CoV-2", the authors have attempted to find nanobodies with broad neutralization potential against different strains of SARS-CoV-2. The authors utilized phage display library with RBD Wuhan Hu-1, RBD Beta, RBD delta, S-protein Omicron as targets to select antigen-specific nanobodies. From their experiments, they found 3 nanobodies that are referred to as PRV, KWL, and SKP that outcompeted ACE2 in binding to the protein target from at least 2 out of the 3 strains tested in their assay. Further, the authors constructed a 3D model to showcase molecular interactions between the RBD regions from 3 SARS-CoV-2 strains and the KWL nanobody. This study contributes to the field of antibody discovery targeting SARS-CoV-2 and also attempts to generate a new strategy for bio panning for the selection of broadly neutralizing nanobodies. The authors must address the following comments for the publication of this manuscript.

1) In the methods section, 2.6, can the authors please mention the duration of the assay and at what time point was recombinant ACE2 added? For a competitive binding assay, the lysates and ACE2 must be added simultaneously, however, the text suggests that ACE2 was added later. Please clarify this assay accordingly.

2) On page 7, second paragraph describes the results of ELISA and the selection criteria for 14 nanobody-expressing phage clones. Please add the data for this paragraph and generate a new figure or add a panel as part of Figure 1.

3) The authors mention that 14 nanobodies had specific interactions with recombinant trimers and those were cloned into a pET21 vector. However, they moved forward with only 4 nanobodies. Did they not get sufficient expression from the remaining 9 nanobodies and have they planned to follow up on these with further optimization? 

4) Between figure 3 and table 1, did the authors expect similar results? Successful competition with ACE2 should be a reflection of the neutralization potential of the nanobodies. In case of PRV, it inhibits binding of ACE-2 to Wuhan Hu-1 but not to Delta, however, the neutralization titre for both strains are 640. Similarly, KWL competes with ACE2 in binding with Wuhan Hu-1 but not Omicron, however, the neutralization titre is 80 for both. Can the authors please explain this difference and if there are any technical reasons for this discrepancy?

5) For figure 4, I suggest that the authors add the contact zone between the RBD and ACE-2 with molecular interactions. The readers will be able to understand the competitive nature of these interactions better with this small addition.

6) The authors have mentioned that 32 phage clones had a stop codon or frame shift and therefore were not chosen for the study. How can these defective sequences get selected if the phage clones are unable to produce a functional full length VHH? 

7) Is it feasible to perform a sequence alignment of the RBD regions of different SARS-Cov-2 strains and identify conserved regions and utilize those peptides as targets to select for broadly neutralizing nanobodies? This will restrict their search criteria and eliminate single-variant specific nanobodies. In their work, the authors observed accumulation of delta-specific nanobodies because that was part of the 3rd bio panning approach and they have mentioned in their discussion that each new target may be selecting for target-specific nanobodies and nanobodies specific to prior targets may be lost during this selection. What alternative method can be used to prevent this?

Some minor comments about the editing in the paper are as below. 

8) In the introduction, there are 3 paragraphs explaining studies from references 26, 27, 28, 9, and 29. This can be shortened to 1 paragraph.

9) Methods 2.2, line 5, "adsorption buffer".

10) Page 9, line 2, the authors should mention only Wuhan and Omicron because delta-ACE2 interaction is not being inhibited by PRV.

Round 2

Reviewer 1 Report

Comments and Suggestions for Authors

The reviewer appreciates the consideration given to the previous suggestions. I believe the changes made have notably improved the robustness of the manuscript. Particularly the clear rationale for the panning strategy is appreciated. A couple of further very minor suggestions would be as follows:

  • Could statistical analysis be conducted to show whether the responses seen in Fig 3 have statistical significance?
  • Could the authors provide a statement towards the end of the manuscript detailing prospective future studies to further this body of work including the afore mentioned epitope binning and binding kinetic studies?
  • Could the authors add text within the manuscript explaining why ELISA against RBD Beta was not conducted?
